# Influence of Nominal Maximum Aggregate Size and Aggregate Gradation on Pore Characteristics of Porous Asphalt Concrete

**DOI:** 10.3390/ma13061355

**Published:** 2020-03-17

**Authors:** Wenke Huang, Xu Cai, Xiang Li, Wentian Cui, Kuanghuai Wu

**Affiliations:** 1School of Civil Engineering, Guangzhou University, Guangzhou 510006, China; h.wenke@gzhu.edu.cn (W.H.); cx_caixu@163.com (X.C.); wentiancui1@163.com (W.C.); 2School of Civil Engineering, Chongqing Jiaotong University, Chongqing 400074, China; lixianghaoyun@163.com

**Keywords:** aggregate size and gradation, pore characteristics, porous asphalt concrete, connectivity, x-ray computer tomography

## Abstract

Porous asphalt concrete (PAC) has been used to improve the traffic conditions in rainy weather due to its high porosity. Aggregate size and gradation have great impact on the connected pore structure, which ultimately affects the permeability of porous asphalt concrete. In this paper, the topological properties of connective pores including pore area, pore circularity, equivalent pore diameter, and void network of porous asphalt concrete with different nominal maximum aggregate sizes and gradations were analyzed using x-ray computer tomography scans and the image processing technique. It was observed that the maximum aggregate sizes will not have significant effect on the percentage of connected pores to total pores for porous asphalt concrete. Furthermore, the percentage of connected pores to total pores is related to the air void content, but for PAC-13 with 20% target air void content or above, the connectivity does not seem to have a sharp increase. Additionally, porous asphalt concrete with a smaller nominal particle size or lower target air void content seems to generate a more concentrated distribution of *Eqdiameter*. Moreover, pore circularities for porous asphalt concrete with a maximum aggregate size of 10 mm or above are independent of maximum aggregate sizes. Air void contents ranging from 16% to 21% do not have a significant effect on the voids’ circularity. Furthermore, the branching nodes in porous asphalt concrete with a smaller nominal maximum aggregate size or lower target air void content have a more uniform spatial distribution. However, the percentage of cross-linked number to total node raises as the nominal maximum aggregate size or target air void content increases.

## 1. Introduction

Porous asphalt concrete (PAC) or open graded friction course (OGFC), as an environmentally friendly road material due to its noise reduction properties, drainage, and improved traffic conditions in rainy weather, has attracted continuous attention in recent years [1,2,3,4,5]. As a complex polymeric porous media, PAC is generally considered to be composed of aggregate, asphalt mastic, and large porosity, which usually has a void content of 14%–31% [6]. Due to higher proportions of porosity, rainwater can quickly infiltrate underground through porous asphalt concrete to prevent aquaplaning on the road surface and improving visibility.

Air voids have a significant influence on the hydraulic conductivity of PAC [7]. Therefore, an number of experimental researches related to porous asphalt have been conducted to analyze the influence of void content on permeability. Suresha [8] carried out permeability tests using the falling-head concept on cylindrical Particle Flow Code (PFC) specimens and concluded that the relationship between the permeability and effective air voids followed a power model. Hamzah [9] conducted permeability tests on porous asphalt with a falling head water permeameter and found out that when the asphalt binder crept, the asphalt binder filled up the empty spaces in the mix air voids, which disrupted void continuity, which translated into the observed reduction in the coefficient of permeability. Chen [10] used a customized testing method to investigate the permeability of the OGFC mixture. Results showed that the OGFC mixture is affected by rainfall intensity and transverse cross slope in addition to the air voids. These factors affect the effective air voids available for water flow and thus water saturation level in the OGFC mixture.

However, these water permeability experiments cannot provide a clear insight into the internal structure of PAC. Thanks to computer tomography (CT) and image processing technologies, the internal structure of asphalt concrete can be easily obtained and it is possible to analyze the characteristics of asphalt concrete in the microstructural domain [11]. Jerjen [12] investigated the water evaporation rate in a sample of porous asphalt concrete by means of x-ray micro computed tomography and found that a qualitative inspection of the pore network allowed a tentative link between the sudden acceleration of evaporation to the disappearance of water lids that were clogging pores. Zhang [13] utilized high-resolution industrial computed tomography (CT) scans to perform on the PA concrete cores that were obtained from the field trial sections to detect the distribution of the rejuvenation products.

In fact, permeability is actually a function of the microstructure of pores in PAC, especially the interconnected pores. Many researchers have paid attention to the pore structure parameters such as porosity, tortuosity, and pore sizes by applying the x-ray computed tomography scans. Alber [14,15] analyzed the characteristic parameters of the air void and skeleton structures and influence of soiling phenomena on air–void microstructures of the porous asphalt through the CT scanning technique. Mahmud [16] used a non-destructive method (CT method) employing virtual cut section to analyze the content, number, shape, and size of voids of laboratory-fabricated porous asphalt. Zhao [17] used CT scanning technology to distinguish the valid and invalid interconnected pores in order to evaluate the distribution of various types of pores, and finally proposed a new evaluation index of pore distribution. Yu [18] identified the pore characteristics in pervious concrete by 2D/3D CT images and investigated the relationship between pore characteristics and permeability. These studies showed the feasibility of combining the computer tomography (CT) method with image processing technology to capture the pore characteristics of porous material. 

While these studies are limited to the effect of air void topology on the hydraulic conductivity of porous asphalt concrete, the relationship between all of these topological parameters and nominal maximum aggregate size and aggregate gradation is not clear yet. Some efforts have been made to investigate the influence of aggregate type and size on the compressive strength of cement-based previous concrete. Ćosić [19] investigated the influence of aggregate type and size on the properties of cement pervious concrete with five different concrete mixtures. Yu [20] studied the relationship between aggregate size, compressive strength, pore structure, cementitious paste thickness, and then analyzed the influence of aggregate size on the compressive strength of pervious concrete. Zhong [21] carried out linking matrix strength, total porosity, aggregate size, and mean pore size of pervious concrete to its compressive strength.

However, few studies have focused on the effect of porous asphalt concrete with nominal maximum aggregate size and aggregate gradation on the pore characteristics from a microstructural perspective. Generally, coarse aggregates (larger than 2.36 mm) are used to constitute the skeleton structure of the porous asphalt concrete and the asphalt mastic including asphalt binder, mineral fillers, and fine aggregates fills the pores formed by coarse aggregates. In this case, the aggregate sizes and the aggregate gradations are critical to producing a mixture that will meet traffic loading and the specific runoff.

## 2. Objectives

Motivated by the aforementioned limitations, the objective of this study was to characterize the connective pore system of porous asphalt concrete with different nominal maximum aggregate sizes and aggregate gradations. To achieve this objective, five kinds of porous asphalt concrete specimens with different nominal maximum aggregate sizes and target air void contents were scanned by industrial CT, and the CT scan results were used to investigate the relationship between the pore space topological properties and aggregate gradations with accessible parameters in this study.

## 3. Materials and Methods 

### 3.1. Materials and Sample Preparation

Aggregate gradation has a great impact on the structural and functional performance of porous asphalt mixture required for a specific application [22,23]. In order to investigate the effect of the nominal maximum aggregate sizes and aggregate gradations on air void properties for porous asphalt concrete, five kinds of commonly-used porous asphalt concretes (PAC-5, PAC-10, PAC-13(1), PAC-13(2), and PAC-13(3)) were designed by the gyratory compaction method. As presented in Table 1, PAC-5, PAC-10, and PAC-13(1) had the same target air void content, which was used to specify the influence of the maximum aggregate sizes on the air void properties. 

Sieves of 2.36 mm and 4.75 mm are usually treated as critical sieves that have a great influence on the air void volume in the mixture [24,25]. Therefore, in order to investigate the effect of gradations on air void properties, three aggregate gradations (PAC-13(1), PAC-13(2), and PAC-13(3)) were designed by changing the percentage passing of aggregate sizes of 2.36 mm and 4.75 mm.

Crushed granite stones were used as coarse and fine aggregates in this study and high-viscosity modified asphalt was selected as the binder. Tests were conducted according to the Chinese specifications (JTG E42-2005 and JTG E20-2011). The detailed properties of the materials used are provided in Table 2. The mixture samples were prepared in accordance with ASTM D 7064. The specimens that were 150 mm in diameter and 100 mm in height were produced using the Superpave gyratory compactor with 50 gyrations. Each of the specimens were compacted at a vertical pressure of 600 kPa and a gyration rate of 30 rpm to achieve the target air void contents of 20%, 20%, 20%, 18%, and 16%, respectively. Then, the specimens were cut into cylinders with a diameter of 100 mm and height of 70 mm due to the limitations of the x-ray scanner (YXLON Compact-225, Hamburg, Germany).

### 3.2. X-Ray Computer Tomography (CT) Scanning and Digital Image Segmentation

In this study, a YXLON Compact-225 X-ray scanner was used to obtain the detailed microscopic structure of the specimens. The highest spatial resolution of the YXLON Compact-225 X-ray scanner can reach 20 microns. We collected a piece of image every 0.1 mm along the specimen height direction of the specimens by industrial CT. The pixel size of the CT slice was 1500 × 1458 and the resolution obtained from the x-ray CT scan was 0.12 mm/pixel. A total of 700 slices of each specimen with the dimensions of 100 mm (diameter) × 70 mm (height) were obtained. In order to eliminate the edge artifacts of the CT slices that affect the accuracy of air void measurements, the slices were cropped to a region of interest (ROI) of 90 mm in diameter and 60 mm in height. Figure 1 shows typical original slices from five kinds of PAC specimens and cropped slices from them.

The CT slices had different grayscale intensities between 0 and 255, where denser materials had a higher intensity. Based on the different gray values, the image processing method was applied to distinguish different materials among the aggregates, asphalt mastic, and air voids. The results of the air void phase in a typical slice of PAC-10 is illustrated in Figure 2a. Volume rendering reconstruction of the air void can be generated by stacking every segmented slice in the AVIZO software (AVIZO 9.0.1, Thermo Fisher Scientific Inc., Waltham, MA, USA) as shown in Figure 2b.

It is known that the isolated pores in the specimen do not allow for the evaluation of the permeability of PAC. As a result, it is important to distinguish the interconnected pores in the specimen before statistically analyzing the air void properties. There are three types of connected domains of pores (6-connected, 18-connected, and 26-connected) in the three-dimensional volume render digital sample, as shown in Figure 3. The rule for 6, 18, and 26 connectivity can be defined as: voxels are connected if their faces touch, voxels are nonconnected if their faces or edges touch, and voxels are connected if their faces, edges, or corners touch, respectively. In this paper, the 6-connected method was used in the process of detecting connected pores in the PAC digital samples and Avizo software was applied for this purpose. Typical visualization of connectivity and disconnectivity of air voids in PAC-10 are illustrated in Figure 4.

### 3.3. Characteristics of Air Void

Air void properties of each CT slice cross-section considered in this study included individual void area, pore circularity, and equivalent void diameter. These properties are commonly used to analyze the geometry of porous media [26,27,28,29]. 

#### 3.3.1. Equivalent Void Diameter

Equivalent void diameter is calculated with Equation (1):(1)$$EqDiameter=\sqrt{\frac{4A}{\pi }}$$
where *EqDiameter* is the void equivalent diameter (mm) and *A* is the pore area (mm^2^).

#### 3.3.2. Pore Circularity

The pore circularity is defined as Equation (2), in which the value of 1.0 indicates a perfect circle and the value of 0.0 means otherwise. High pore circularity facilitates water transport in the PAC with the increased accessibility.
(2)$$C=\frac{4\pi A}{L^{2}}$$
where *C* is the pore circularity and ranges between 0 and 1; *A* is the pore area (mm^2^); and *L* is the pore perimeter (mm).

## 4. Results and Discussion

### 4.1. Pore Distribution

Pore distribution is an important indicator as it can describe the pore characteristics of porous asphalt concrete. The area of pores and percentages of connected pores to total pores along specimen height direction in this study were selected to evaluate the pore distribution of PAC with different nominal maximum aggregate sizes and aggregate gradations.

As shown in Figure 5, by comparing PAC-5, PAC-10, and PAC-13(1), which had the same target air void content, it can be observed that these three types of PAC specimen exhibited similar distribution characteristics along the specimen height direction. In terms of connected pores along the specimen height direction, it can be seen that the PAC specimen with high maximum aggregate sizes tended to have a slightly higher connectivity as presented in Figure 6 and Table 3. However, the mean percentages of connected pores to total pores for PAC-5, PAC-10, and PAC-13(1) were all above 90%. This is an indication that the maximum aggregate sizes had no significant effect on the percentage of connected pores to total pores of the porous asphalt mixture. This has also been observed by other researchers [30]. One of the main reasons that cause the high-volume connectivity of pores is due to the large porosity in PAC. In the interlocking process of aggregate skeletons in the PAC specimen with a higher porosity, the fine aggregates will fill the voids. While porous asphalt concrete has fewer fine aggregates, which means that the pores created by large aggregates cannot be filled up by the fine aggregates and mineral fillers. On the other hand, in the procedure of voxel connectivity detection, as shown in Figure 3, the dimension of voxel distinguishing the interconnected pores was 0.12 mm in length, width, and height, which means that many voids will be linked by tiny pores. Therefore, the volume of connectivity of the porous asphalt concrete will be high. For PAC-13(1), PAC-13(2), and PAC-13(3) with different target air void contents, the mean percentage of connected pores to total pores increased largely as the target air void content increased, as shown in Figure 5 and Figure 6 and Table 3, which means that the percentage of connected pores is related to the air void content. It also can be seen in Table 3 that the mean percentage of connected pores to total pores of PAC-13(1) was up to 96.2%. In other words, for PAC-13 with a 20% target air void content or above, the connectivity does not seem to have a sharp increase. A similar result was obtained by Aboufoul and Garcia [31]. This is because the PAC specimen with higher air void content had less fine aggregates, filling the voids created by the coarse aggregates. When the porosity of PAC increases to a certain extent (96.2% connectivity for 20% target void in this paper), the connectivity of the PAC specimen will not improve sharply anymore. Based on this, these findings can provide future designers with a reference referring to hydraulic conductivity.

### 4.2. Void Dimensional Property

To predict the influence of nominal maximum aggregate sizes and aggregate gradations on air void characteristics, pore circularity and equivalent void diameter were selected for discussion in this section.

The equivalent void diameter of every pore in each CT slice was calculated through Equation (1). As shown in Figure 7a, the percentage distributions of pore number in different equivalent void diameter ranges were highly consistent. The diameter range corresponding to the peak point percentage for PAC-5, PAC-10, PAC-13(1), PAC-13(2), and PAC-13(3) was 0.4 to 0.8 mm. For porous asphalt concrete (PAC-5, PAC-10, and PAC-13(1)) with different nominal maximum aggregate sizes, the peak point percentages reached 25.0%, 14.7%, and 11.8%, respectively. That is, porous asphalt concrete with a smaller nominal particle size tends to have a narrower distribution of *Eqdiameter*. This is mainly because porous asphalt concrete with a smaller nominal maximum aggregate size has a uniform particle distribution. Therefore, the PAC specimen with a smaller nominal maximum aggregate size has a more uniform pore size, resulting in a more concentrated pore distribution. To further clarify this observation, the cumulative percentage of pore number vs. range of *Eqdiameter* was drawn in Figure 7b. As shown in Figure 7b, nearly 90% of the pore diameters for PAC-5 were located between 0 and 1.4 mm, but for PAC-10 and PAC-13(1), the cumulative percentage of pore number fell in the range of 0 to 2.8 mm and 0 to 3.6 mm, respectively. Another important observation was that the peak point percentages of the porous asphalt concrete were affected by the target air void content. That is, pore size distribution was narrower as the target air void content increased, as shown in Figure 7a. Figure 7b can also provide a direct visualization for the pore size distribution characteristic affected by the target air void content by taking into account the cumulative percentage of pore number, where the cumulative percentage of pore number fell in the range of 0 to 2.8 mm and 0 to 3.6 mm for PAC-13(1), PAC-13(2), and PAC-13(3), respectively. A total of 90% of the pore diameters for PAC-13(1), PAC-13(2), and PAC-13(3) were located in the pore size category from 0 to 3.6 mm, 0 to 2.8 mm, and 0 to 2.6 mm, respectively. The observations above denote that porous asphalt concrete with a smaller nominal particle size or lower target air void content seems to generate a more concentrated distribution of *Eqdiameter*.

To evaluate the water transport potential of the PAC with different nominal particle size or target air void content, the pore circularity distributions calculated with Equation (2) were investigated by counting the number of pores in each category. The pore circularity distributions along the specimen height direction presented in Figure 8a shows that PAC-10, PAC-13(1), PAC-13(2), and PAC-13(3) had a similar category of pore circularity ranging from 0.55 to 0.65, which was very different from that of PAC-5 with a range of 0.7 to 0.75. It was also found that the mean pore circularity of PAC-10, PAC-13(1), PAC-13(2), and PAC-13(3) were very close to each other while distinctly different from PAC-5, as shown in Table 4. Researchers [30] have reported that void circularities of porous asphalt concrete with 14 mm, 20 mm, and 28 mm maximum aggregate sizes for 17% air void content were 0.54, 0.57, and 0.54, respectively, which did not sharply change. This means that the pore circularities for test samples with the maximum aggregate size of 10 mm or above are independent of the maximum aggregate size compared with the results analysis in this study. Furthermore, void circularities of porous asphalt concrete with a 20 mm maximum aggregate size for 13%, 17%, 21%, and 26% air void contents were 0.59, 0.57, 0.55, and 0.49, respectively [30]. This means that the void circularity of porous asphalt concrete with air void contents from 16% to 21% had a small range of 0.55 to 0.62, as seen in the combined results in Table 4 and [30]. This finding implies that air void contents ranging from 16% to 21% do not have a significant effect on the voids’ circularity. However, additional specimens with air void contents above 21% should be investigated to further discuss the influence of air void content on void circularity in the future. Additionally, Figure 8b illustrates the percentage of pore number in the range of pore circularity for the five types of specimens. It is noteworthy that PAC-10, PAC-13(1), PAC-13(2), and PAC-13(3) had a consistent distribution while the distribution for PAC-5 was a little different. This implies that PAC-5 tends to have a narrower pore circularity distribution, which is similar to that of an equivalent void diameter. The peak point percentage of pore number for the five kinds of specimens fell in the category of 0.8 to 0.9. However, the percentage of pore number at the peak point for PAC-5 was about 30.9%, sharply higher than that of the other four specimens. This finding implies that PAC-5 tends to have more regular shapes.

### 4.3. A New Proposed Void Connnectivity Index: Cross-Linked Number

Luo [32] found that the number of connected paths was the best predictor for the hydraulic conductivity of soils. It can be drawn that the permeability efficiency of two cross-linked water flow paths is higher than that of two non-cross-linked water flow paths because more crosslinking means more flow paths for water to transit, as shown in Figure 9. Therefore, the number of node including the terminal node and branching node, as shown in Figure 9a, can represent the degree of cross-linking for water flow paths. However, a subset of the voids are connected with the outside, but make no contribution to the permeability of the mixture [17]. In this paper, the branching node was selected to evaluate the potential permeability of the porous asphalt mixture.

Spatial distribution of five kinds of porous asphalt mixture specimens can be found in Figure 10. The distribution of branching nodes in the specimen with a smaller nominal maximum aggregate size had a more uniform spatial distribution, as shown in Figure 10b,d,f. In other words, with the decrease in the maximum aggregate size, the cross-linked points gradually converge along the column center of the specimen and form a large cluster at the center of the specimen along the height direction. This is mainly because aggregates in the specimen with a smaller aggregate size are easy to rotate during compaction, so that the pore distribution is well-distributed. For PAC-5, PAC-10, and PAC-13(1), which had the same target air void content, the percentage of cross-linked number to total node (including terminal nodes and branching nodes) of 66.1%, 77.0%, and 79.0%, respectively, rises as the nominal maximum aggregate sizes increase, as illustrated in Table 5. In turn, cross-linked points are opposite in distribution with the decrease of the target air void content due to its larger content of fine aggregates, as shown in Figure 10f,h,j. While the percentage of the cross-linked number to total node for PAC-13(1), PAC-13(2), and PAC-13(3) were 79.0%, 71.5% and 69.5%, respectively, were the opposite, with the decrease in the target air void content due to its larger content of fine aggregates in lower air void content specimens. This finding indicates that porous asphalt concrete with a larger nominal maximum aggregate size or higher air void content seems to have a greater inter-connectivity. Though the porous asphalt concrete with a lower target air void content (ranging from 16% to 20%) possesses a uniformed pore distribution, more specimens with different target air void contents need to be conducted to investigate the distribution characteristics of branching nodes.

## 5. Conclusions

This paper investigated the microstructural connective pore properties of porous asphalt concrete with different nominal maximum aggregate sizes and gradations using x-ray computer tomography scans and an image processing technique. The pore characteristics including pore area, pore circularity, equivalent pore diameter, and void network of porous asphalt concrete were studied in order to find the relationship between aggregate sizes, aggregate gradations, and the permeability of PAC. According to the results, the main conclusions can be given as follows:1)PAC specimens with a high maximum aggregate size tend to have a slightly higher connectivity. However, the mean percentages of connected pores to total pores for porous asphalt concrete with the same target air void content were all above 90%, which means that the maximum aggregate size will not have a significant effect on the percentage of connected pores to total pores for porous asphalt concrete. Furthermore, the percentage of connected pores is related to the air void content, but for PAC-13 with a 20% target air void content or above, the connectivity does not seem to have a sharp increase.2)Porous asphalt concrete with a smaller nominal particle size or lower target air void content seems to generate a more concentrated distribution of *Eqdiameter*. Furthermore, pore circularities for porous asphalt concrete with a maximum aggregate size of 10 mm or above are independent of maximum aggregate sizes, and air void contents ranging from 16% to 21% do not have a significant effect on the voids’ circularity.3)The branching nodes in porous asphalt concrete with a smaller nominal maximum aggregate size or lower target air void content seem to have a more uniform spatial distribution. However, the percentage of cross-linked number to total node rises as the nominal maximum aggregate size or target air void content increases. This finding indicates that porous asphalt concrete with larger nominal maximum aggregate sizes or target air void content seems to have a greater inter-connectivity.

## 6. Further Research

Additional specimens with a wide range of nominal particle sizes and air void contents should be investigated.

## Figures and Tables

**Figure 1 materials-13-01355-f001:**
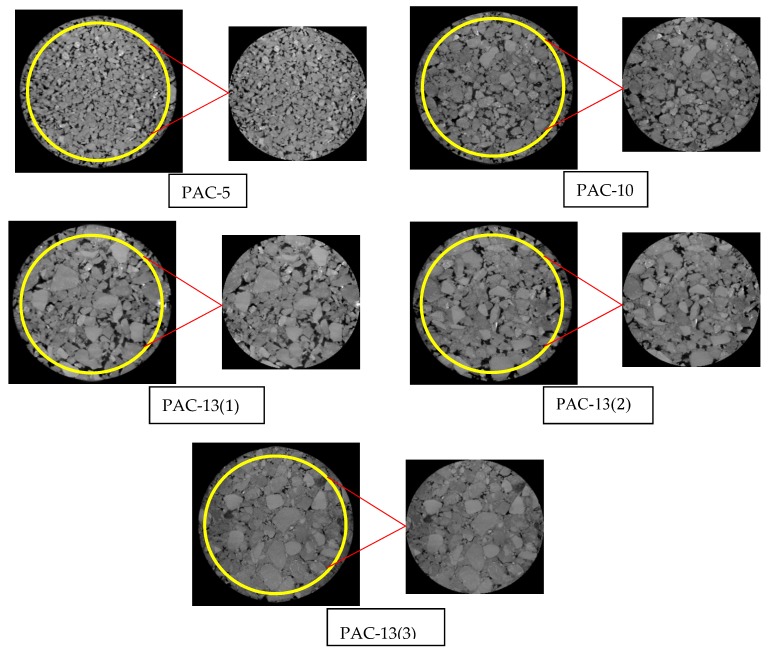
Illustration of cropping a region of interest from the original slices.

**Figure 2 materials-13-01355-f002:**
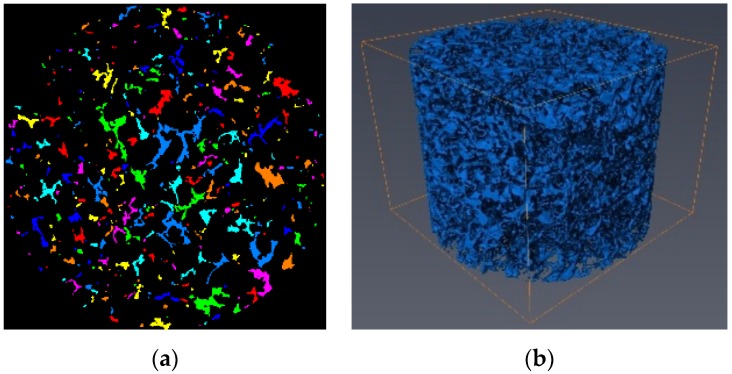
(**a**) Air void phase in a typical slice. (**b**) Volume rendering reconstruction.

**Figure 3 materials-13-01355-f003:**
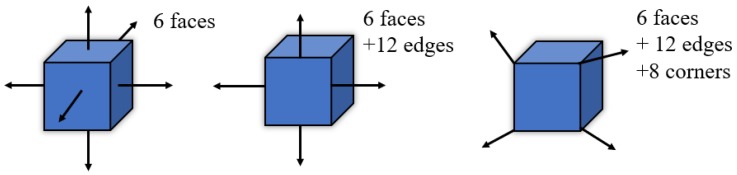
Voxel connectivity.

**Figure 4 materials-13-01355-f004:**
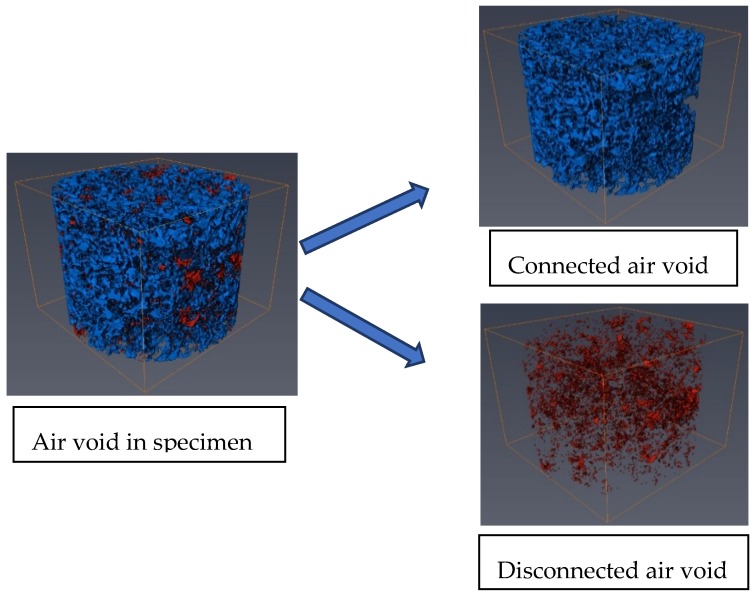
Visualization of connectivity and disconnectivity of the air void.

**Figure 5 materials-13-01355-f005:**
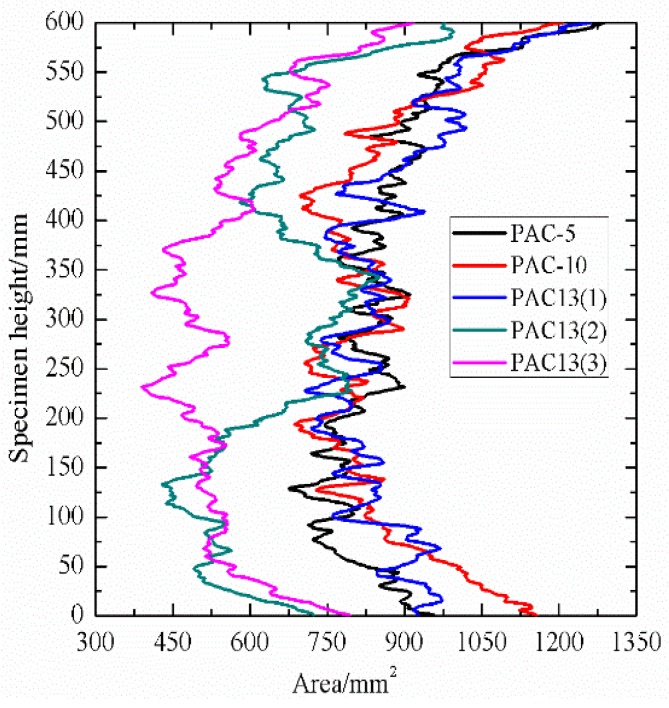
Area of pores along the specimen height direction.

**Figure 6 materials-13-01355-f006:**
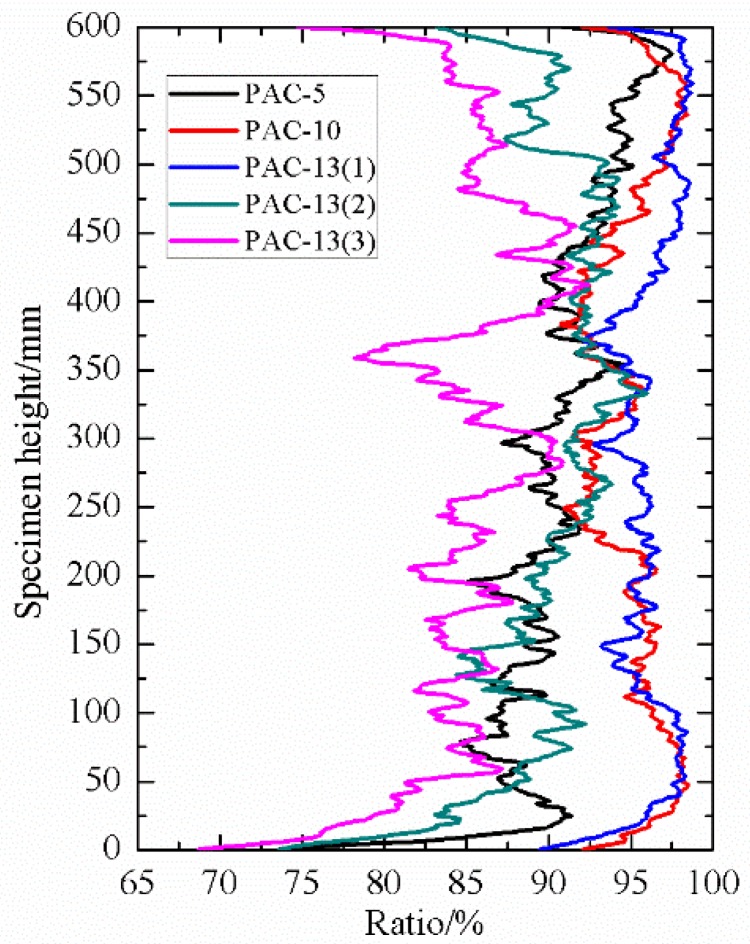
Percentage of connected pores to total pores along the specimen height direction.

**Figure 7 materials-13-01355-f007:**
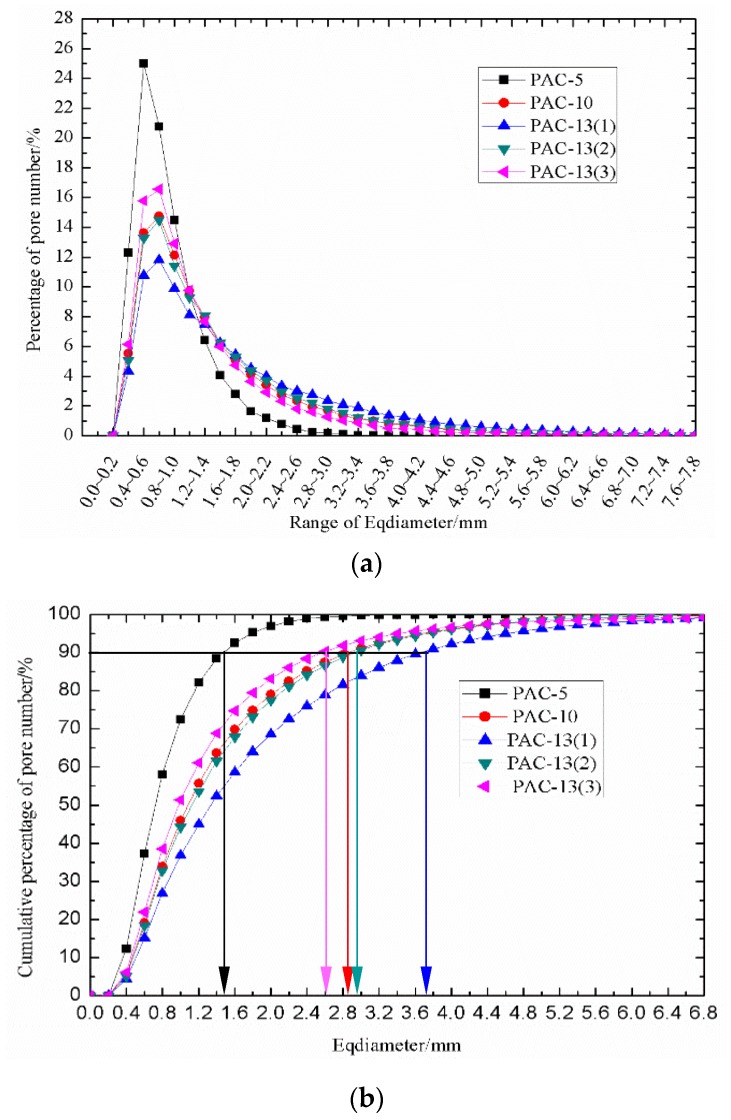
(**a**) Percentage of pore number in the range of *Eqdiameter*. (**b**) Cumulative percentage of pore number in range of *Eqdiameter*.

**Figure 8 materials-13-01355-f008:**
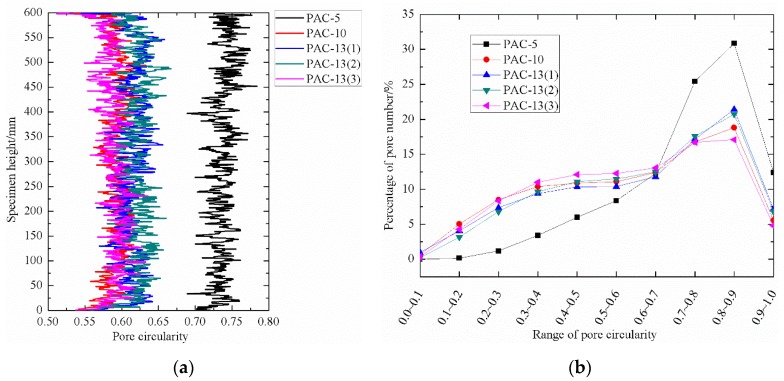
(**a**) Pore circularity distributions along the specimen height direction. (**b**) Percentage of pore number in the range of pore circularity.

**Figure 9 materials-13-01355-f009:**
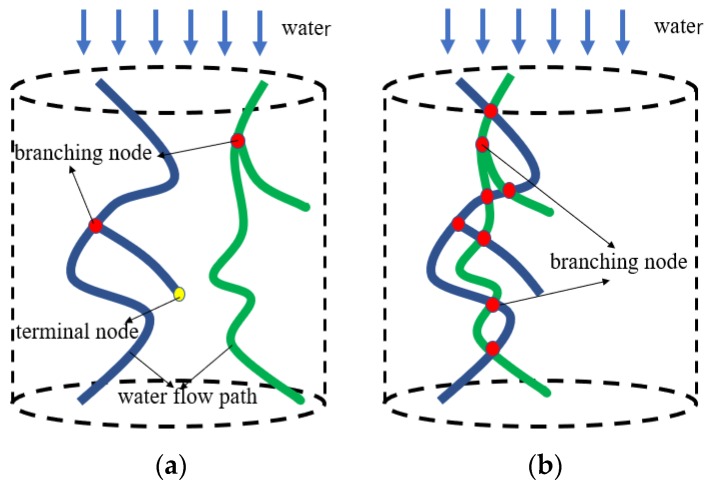
Graphs representing the through-depth water flow path pattern of porous asphalt concrete: (**a**) non-cross-linked; (**b**) cross-linked.

**Figure 10 materials-13-01355-f010:**
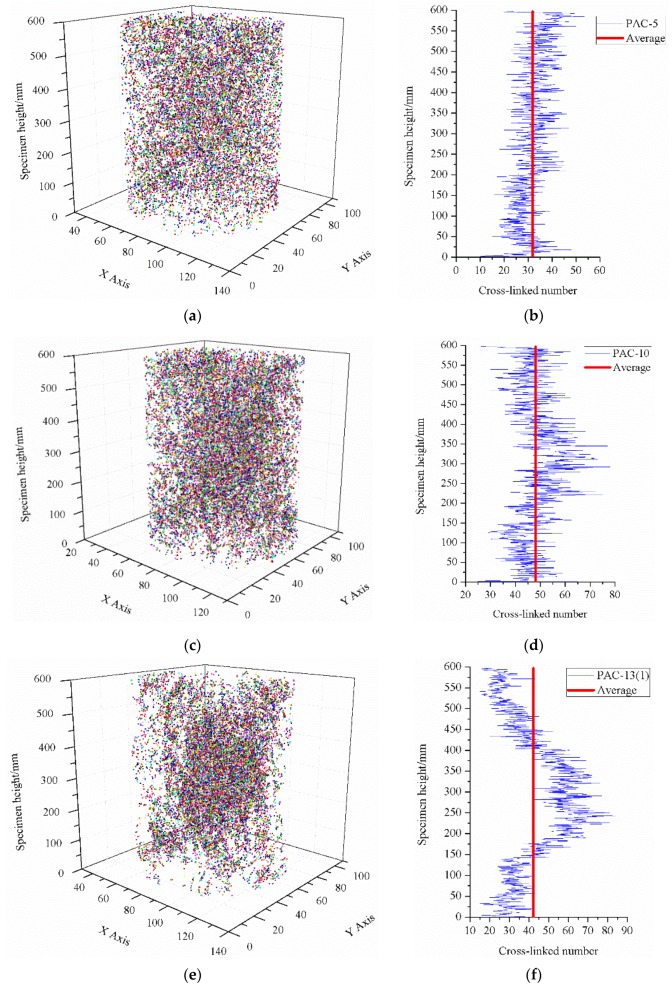
Cross-linked number of interconnected pores along the specimen height direction. (**a**) Cross-linked number spatial distribution of PAC-5; (**b**) Cross-linked number of PAC-5 along the specimen height direction. (**c**) Cross-linked number of the spatial distribution of PAC-10; (**d**) Cross-linked number of PAC-10 along the specimen height direction; (**e**) Cross-linked number of the spatial distribution of PAC-13(1); (**f**) Cross-linked number of PAC-13(1) along the specimen height direction; (**g**) Cross-linked number of the spatial distribution of PAC-13(2); (**h**) Cross-linked number of PAC-13(2) along the specimen height direction (**i**) Cross-linked number spatial distribution of PAC-13(3); (**j**) Cross-linked number of PAC-13(3) along the specimen height direction.

**Table 1 materials-13-01355-t001:** Gradations of porous asphalt concrete (PAC) with different maximum aggregate sizes and air void contents.

Sieve Size (mm)	Percentage Passing (%)				
	PAC-5	PAC-10	PAC-13(1)	PAC-13(2)	PAC-13(3)
16	–	–	100.0	100.0	100.0
13.2	–	100.0	87.0	87.0	87.0
9.5	100.0	84.7	63.7	63.7	63.7
4.75	90.0	22.8	22.0	22.0	28.3
2.36	21.1	15.6	16.5	19.3	22.1
1.18	20.0	12.6	14.0	14.0	14.0
0.6	15.5	9.4	9.2	10.2	10.2
0.3	11.9	6.5	6.3	7.2	7.2
0.15	9.1	5.1	4.5	5.2	5.2
0.075	7.0	4.6	4.0	4.7	4.7
Asphalt binder (%)	6.3	5.7	5.67	5.8	5.9
Target air void content (%)	20	20	20	18	16

**Table 2 materials-13-01355-t002:** Properties of the materials.

Materials	Physical Properties	Unit	Test Results	Test Method
Coarse Aggregate	Relative Apparent Density	–	2.601	T0304
	Water Absorption	%	0.93	T0304
	Aggregate Crushed Value	%	18.25	T0316
Fine Aggregate	Relative Apparent Density	–	2.653	T0328
	Clay Content	%	1.2	T0333
	Sand Equivalent	%	75	T0334
Mineral Filler	Relative Apparent Density	–	2.606	T0352
	Moisture Content	%	0.8	T0332
High-viscosity Modified Asphalt	Softening Point	°C	81.8	T0606
	Penetration at 25 °C	0.1 mm	43.1	T0604
	Viscosity at 135 °C	Pa.s	4.72	T0619

**Table 3 materials-13-01355-t003:** Pore distribution properties.

Aggregate Gradation	PAC-5	PAC-10	PAC-13(1)	PAC-13(2)	PAC-13(3)
Target air void content/%	20	20	20	18	16
Mean percentage of connected pore to total pore /%	90.6	94.1	96.2	90.1	85.2

**Table 4 materials-13-01355-t004:** Mean pore circularity.

Aggregate gradation	PAC-5	PAC-10	PAC-13(1)	PAC-13(2)	PAC-13(3)
Mean pore circularity	0.74	0.59	0.62	0.62	0.59

**Table 5 materials-13-01355-t005:** Cross-linked number properties.

Aggregate gradation	PAC-5	PAC-10	PAC-13(1)	PAC-13(2)	PAC-13(3)
Percentage of cross-linked number to total Node/%	66.1	77.0	79.0	71.5	69.5

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
