# Peer review of "Influence of Nominal Maximum Aggregate Size and Aggregate Gradation on Pore Characteristics of Porous Asphalt Concrete"

_materials, 2020, doi:10.3390/ma13061355_

Round 1

Reviewer 1 Report

Porous asphalt concrete (PAC) is useful in improving the traffic conditions in rainy weather. A larger aggregate size is definitely can improve the traffic safety. You have done a good job in proving your case.

Author Response

Responses have be attached as a PDF file.

Reviewer 2 Report

The article entitled "Influence of nominal maximum aggregate size and aggregate gradation on pore characteristics of porous asphalt concrete" is an interesting material in the field of computed tomography on the distribution of porous asphalt pores. Present literature has a limited collection of articles on this subject. The article may also be a good starting point for further, more advanced analyzes of mechanical properties of this type of mixtures. In the reviewer's opinion, the article was written in the correct English language. Here are some remarks:

  1. In Table 2 neither the standards according to which the tests were carried out were given, nor units of the examined features were given.

  1. Whether the applied principle of pore domain connections (lines 122 to 126) has already been confirmed in the literature?

  1. Quantitative analysis of test results included in point 3 should be extended. The authors limited themselves only to visual comparison or argued based on the average values. Observing the results in fig. 7a, the results distribution does not authorize to evaluate the test results using mean. In this case the median is proper estimator . Therefore, pore distribution analysis should start with matching the probability distribution of results (it does not look as normal distribution). This will be particularly useful and reliable when choosing significance tests. It will also allow to build a potential statistical generalized model that correctly shows the correlation between the features in Table 3. Therefore, the presented analysis of results is a declaration of results and not analysis. Accordingly, this article must be supplemented with this correct quantitative analysis.

Author Response

Response has been attached a PDF file.

Reviewer 3 Report

This paper (Influence of nominal maximum aggregate size and aggregate gradation on pore characteristics of porous asphalt concrete) presents interesting results but needs a thorough revision before being considered for publication. Some sections need to be completely rewritten, like the Introduction, literature review and Discussion.

Introduction: The theoretical, analytical and standard approaches should be discussed.

The novelties have to be outlined. It has to be completely rewritten so that the focus of the work and its innovative content can be really appreciated.
Literature review: The Literature review is now a mere list of information but the authors have to provide their own "unifrying" view and not only citing previous work.

In addition, in a quick search I found a number of papers on this topic that you did not cite. I am listing them here, please consider them in the literature review and in the interpretation of the results:

  1. Drying of Porous Asphalt Concrete Investigated by X-Ray Computed Tomography, Physics Procedia Volume 692015, Pages 451-456, I. Jerjen, L. D. Poulikakos, M. Plamondon, Ph. Schuetz, A. Flisch
  2. Using high-resolution industrial CT scan to detect the distribution of rejuvenation products in porous asphalt concrete, Construction and Building Materials, Volume 10015 December 2015, Pages 1-10, Y. Zhang, W. Verwaal, M. F. C. van de Ven, A. A. A. Molenaar, S. P. Wu
  3. A review of X-ray computed tomography of concrete and asphalt construction materials, Construction and Building Materials, Volume 19928 February 2019Pages 637-651, Anton du Plessis, William P. Boshoff

Results and discussion: The paper presents a few amount of results from unusual experiments but without a theoretical and practical approach.

Conclusions: The discussion about technological benefit have to be separated in the article according points of conclusions. The analysis of the results is quite basic and deserves better and deeper processing.

Author Response

Response has been attached a PDF file.

Reviewer 4 Report

The paper is well written. However, few corrections are needed. There is a need for further discussion of the results. The introduction needs more literature review. The English writing is fine but it needs minor corrections. 

Author Response

Response has been attached a PDF file.

Reviewer 5 Report

The subject matter is interesting, and the research is planned and performed correctly. However, according to the reviewer, the article contains too few new results to be published in the renowned journal Materials. The introduction is very concise and contains little information. Similarly, the description of the research, sometimes there are 2-3 lines of text which is not enough. X-ray CT scanning is of course modern and interesting, but the analysis of the obtained results is very cursory and no comparison with the results obtained by other authors. Only 5 series of specimens were made for testing. Most drawings have too small descriptions and are illegible. Conclusions are quite general and not all of them result from the presented research results. The article could be published, but this requires additional test results, thorough rewriting and completion of the text. In particular the introduction and discussion of the results.

Author Response

Response has been attached a PDF file.

Round 2

Reviewer 5 Report

The authors have made many amendments to the text that have increased the value of the article. In its current form it can be published after minor linguistic corrections.